# Cryo-EM structures of the TTYH family reveal a novel architecture for lipid interactions

Anastasiia Sukalskaia [1], Monique S. Straub[1], Dawid Deneka[1], Marta Sawicka [1✉] & Raimund Dutzler [1✉]

The Tweety homologs (TTYHs) are members of a conserved family of eukaryotic membrane proteins that are abundant in the brain. The three human paralogs were assigned to function as anion channels that are either activated by $Ca^{2+}$ or cell swelling. To uncover their unknown architecture and its relationship to function, we have determined the structures of human TTYH1–3 by cryo-electron microscopy. All structures display equivalent features of a dimeric membrane protein that contains five transmembrane segments and an extended extracellular domain. As none of the proteins shows attributes reminiscent of an anion channel, we revisited functional experiments and did not find any indication of ion conduction. Instead, we find density in an extended hydrophobic pocket contained in the extracellular domain that emerges from the lipid bilayer, which suggests a role of TTYH proteins in the interaction with lipid-like compounds residing in the membrane.

---

[1] Department of Biochemistry, University of Zurich, Zurich, Switzerland. ✉email: m.sawicka@bioc.uzh.ch; dutzler@bioc.uzh.ch

The Tweety homologs (TTYHs) constitute a conserved family of membrane proteins that are ubiquitously expressed in multicellular eukaryotes. The founding member Tweety was originally identified as part of the flightless locus in drosophila[1]. Its genetic knockout in the fly did not result in a pronounced phenotype, which was partly ascribed to a potential functional complementation by another paralog found in drosophila, and thus did not provide insight into its physiological role[1,2]. In humans, the family contains three paralogs termed TTYH1, 2 and 3 (refs. [2–4]). These proteins are widely distributed in the body with TTYH1 and 2 being confined to the brain, spinal cord, and testis and TTYH3 showing broad expression patterns[5–8]. TTYH1 is among the most abundant membrane proteins in glia[9] and all three homologs are overexpressed in cancers with TTYH2 being correlated with bad prognostic properties[4,10–13]. An initial phenotype of TTYH1 knockout mice leading to early embryonic lethality[14] could not be confirmed in a subsequent study, which instead correlated the deficiency of TTYH1 with disturbances of the Notch signaling pathway thereby interfering with the differentiation of neuronal stem cells[15].

On a protein level, the three human paralogs range from 450 to 534 amino acids. The length difference can be ascribed to an extended C-terminal region, which was proposed to confer $Ca^{2+}$ sensitivity[3,14]. The conserved N-terminal part of the protein contains several hydrophobic stretches and was predicted to encompass four to six transmembrane segments[16,17].

Early on, human TTYH proteins were associated with chloride channel function. After their overexpression in mammalian cells, a first study assigned all three homologs as chloride channels of large conductance[3,17]. Activation by intracellular $Ca^{2+}$ was detected for TTYH2 and 3 but not for TTYH1[3,17]. A recent study has proposed TTYH1 and its paralogs to act as volume-regulated anion channels (VRACs) in the brain[18,19]. This proposal was founded on the emergence of endogenous currents in response to prolonged swelling of cells and was observed in astrocytes and upon heterologous overexpression of the proteins in mammalian cells[18]. Both studies also claimed to have identified distinct residues that alter ion conduction properties and that were thus suggested to form part of the pore lining[3,18]. A subsequent study has assigned an equivalent function to TTYH1 and 2 in cancer cells[20].

Despite their proposed function as chloride channels, little is known about the structural properties and the oligomeric organization of TTYHs, which are unrelated to any other characterized family of membrane proteins. To overcome this knowledge gap, we have here determined the structures of the three human paralogs by cryo-electron microscopy (cryo-EM). All structures provide equivalent views of a dimeric membrane protein that does not contain characteristic features of an anion channel. After revisiting their proposed functional properties by patch-clamp electrophysiology, we did not observe any evidence of ion conduction mediated by either protein. Instead, the structures show features that hint at a potential involvement of TTYH proteins in the transport, binding, or metabolism of lipids or other membrane-associated compounds.

## Results

**Cryo-EM structures of three human TTYH paralogs**. We were interested in the common architecture of TTYH proteins and its relationship to the postulated function as anion channels and thus decided to study the three human paralogs. On a sequence level, the first 410 residues of all three proteins are highly conserved and the most pronounced difference between paralogs concerns the aforementioned presumably unstructured C-terminus, which

is extended in TTYH2 and 3 (Supplementary Fig. 1a, b). In all three proteins, this region contains a stretch of acidic residues, which was suggested to form a $Ca^{2+}$-binding site. When expressed in HEK293 cells and purified in the detergent glycoldiosgenin (GDN), all three proteins eluted as single peaks on size exclusion chromatography (SEC) at volumes that indicate corresponding oligomeric assemblies of similar size (Supplementary Fig. 1c–e). To obtain a comprehensive overview of TTYH conformations, we have investigated the structural properties of all three paralogs in distinct environments (Supplementary Figs. 2–6 and Table 1): The structures of TTYH2 and 1 were determined in GDN under $Ca^{2+}$-free conditions at 3.3 and 4.0 Å, respectively (Supplementary Figs. 2 and 4). Both proteins display a novel fold with very similar general features. To probe potential consequences of $Ca^{2+}$ binding, we have determined the structure of TTYH3 in presence of $Ca^{2+}$ at 3.2 Å (Supplementary Fig. 5). In this case, we find a large correspondence to the structures obtained in absence of $Ca^{2+}$ without pronounced differences that could be ascribed to the interaction with the divalent cation. Finally, we investigated the impact of a lipid bilayer on the protein and thus reconstituted TTYH2 into lipid nanodiscs and determined its structure at 3.9 Å in a conformation that is indistinguishable from the structure of the same protein in detergent (Supplementary Fig. 3). For all three paralogs, we find an equivalent dimeric organization of subunits that define a novel protein architecture (Fig. 1). For simplicity, we describe these common features based on the TTYH2 structure and emphasize differences to other paralogs wherever relevant. The rectangular cross-section of the membrane-inserted domain of TTYH proteins delineates an elongated dimeric assembly that is 90 Å long and 40 Å wide (Fig. 1a, b). Perpendicular to the membrane, the entire structure is about 120 Å high, excluding contributions of the unstructured intracellular C-terminus (Fig. 1a). A common pronounced feature of TTYH proteins concerns a large, 60 Å long and 55 Å wide extracellular structure, which protrudes 75 Å from the membrane plane and that constitutes the bulk of the dimer interface (Fig. 1a–d). In TTYH2, the interactions between subunits are exclusively mediated by this extracellular component, which buries 1800 Å² of the combined molecular surface in an extended dimer interface. Whereas in both structures of TTYH2 determined in detergent and in lipid nanodiscs, the two transmembrane domains (TMDs) are separated by about 7 Å and thus not in direct contact (Fig. 1a and Supplementary Figs. 2 and 3), they have approached each other in the TTYH1 and 3 structures by mutual rigid body rotations of subunits by 6–7° around an axis located at the extracellular dimer interface (Fig. 1c, d and Supplementary Figs. 4, 5, and 7). The described conformational differences lead to interactions in the transmembrane parts of TTYH1 and 3, which increase the contact area between interacting subunits to 2600 and 3655 Å², respectively.

**Subunit organization**. When viewed perpendicular to the symmetry axis, the dimeric proteins can be divided into distinct structural units according to their disposition with respect to the membrane (Fig. 2a, b). These encompass a cytoplasmic domain composed of a mobile C-terminal region, which is not defined in the cryo-EM density, a TMD, and a glycosylated extracellular domain. The extracellular domain in turn consists of two regions, one being proximal to the membrane (Ex1) and a second, distal region (Ex2), which is involved in inter-subunit interactions (Fig. 2).

On a subunit level, all three paralogs are generally similar, which is reflected in the low root-mean-square deviation of 2.3 and 2.5 Å calculated from superpositions of Cα atoms of TTYH2

**Table 1 Cryo-EM data collection, refinement, and validation statistics.**

| | Dataset 1 Ttyh2 GDN (EMD-13194) (PDB 7P54) | Dataset 2 Ttyh2 ND (EMD-13201) (PDB 7P5M) | Dataset 3 Ttyh3 GDN (EMD-13198) (PDB 7P5C) | Dataset 4 Ttyh1 GDN | Dataset 5 Ttyh1 GDN | Datasets 4 and 5 combined (EMD-13200) (PDB 7S5PJ) |
|---|---|---|---|---|---|---|
| *Data collection and processing* | | | | | | |
| Microscope | FEI Titan Krios | FEI Titan Krios | FEI Titan Krios | FEI Titan Krios | FEI Titan Krios | |
| Camera | Gatan K3 GIF | Gatan K3 GIF | Gatan K3 GIF | Gatan K3 GIF | Gatan K3 GIF | |
| Magnification | 130,000 | 130,000 | 130,000 | 130,000 | 130,000 | |
| Voltage (kV) | 300 | 300 | 300 | 300 | 300 | |
| Electron exposure (e−/Å²) | 61 | 61 | 69.56 | 61 | 61 | |
| Defocus range (μm) | −2.4 to −0.8 | −2.4 to −0.8 | −2.4 to −1.0 | −2.4 to −0.8 | −2.4 to −1.0 | |
| Pixel size (Å)ᵃ | 0.651 (0.326) | 0.651 (0.326) | 0.651 (0.326) | 0.651 (0.326) | 0.651 (0.326) | |
| Initial particle images (no.) | 3,565,080 | 2,682,068 | 1,448,259 | 652,846 | 1,106,223 | |
| Final particle images (no.) | 267,069 | 486,362 | 494,792 | 65,381 | 29,544 | 94,925 |
| Symmetry imposed | C2 | C2 | C2 | C2 | C2 | C2 |
| Map resolution (Å) | 3.3 | 3.9 | 3.2 | 4.3 | 4.8 | 4.0 |
| FSC threshold 0.143 | | | | | | |
| Map resolution range (Å) | 3.1–6.5 | 3.6–6.8 | 3.1–6.6 | | | 3.9–9.2 |
| *Refinement* | | | | | | |
| Model resolution (Å) | 3.52 | 4.0 | 3.2 | | | 4.1 |
| FSC threshold 0.5 | | | | | | |
| Map sharpening *b*-factor (Å²) | −90 | −247.3 | −122 | | | −154.4 |
| Model composition | | | | | | |
| Non-hydrogen atoms | 6246 | 6202 | 6222 | | | 6242 |
| Protein residues | 794 | 788 | 784 | | | 798 |
| Sugar | 12 | 12 | 8 | | | 8 |
| *B* factors (Å²) | | | | | | |
| Protein | 68 | 60 | 51 | | | 70 |
| Sugar | 80 | 67 | 59 | | | 65 |
| R.m.s. deviations | | | | | | |
| Bond lengths (Å) | 0.003 | 0.003 | 0.004 | | | 0.003 |
| Bond angles (°) | 0.511 | 0.554 | 0.578 | | | 0.647 |
| Validation | | | | | | |
| MolProbity score | 1.36 | 1.47 | 1.4 | | | 1.55 |
| Clashscore | 6.58 | 8.8 | 6.92 | | | 8.75 |
| Poor rotamers (%) | 0 | 0 | 0 | | | 0 |
| Ramachandran plot | | | | | | |
| Favored (%) | 98.35 | 99.1 | 97.94 | | | 97.59 |
| Allowed (%) | 1.65 | 0.9 | 2.06 | | | 2.41 |
| Disallowed (%) | 0 | 0 | 0 | | | 0 |

ᵃValues in parentheses indicate the pixel size in super-resolution.

with TTYH1 and 3, respectively (Supplementary Fig. 7a, b). The general topology follows a previous prediction based on the analysis of N-glycosylation sites[16]. Each protein chain consists of five membrane-spanning helices (TM1–TM5) with the N-terminus located on the outside and a C-terminus in the cytoplasm (Fig. 2a and Supplementary Fig. 1a, b). The extended N-terminus preceding the first transmembrane helix (TM1) contains a short helix (N1) that is presumably embedded in the headgroup region (Fig. 2a and Supplementary Fig. 1a, b). In the membrane-inserted TMD, TM1 is found at the periphery of the protein contacting TM4 on the extracellular and TM5 on the intracellular leaflet of the bilayer (Fig. 2a, b). The remaining four helices (TM2–TM5) form a tightly interacting bundle (Fig. 2a, c). The second transmembrane helix (TM2) seamlessly merges into the extracellular helix E1 and spans the entire structure (Fig. 2a and Supplementary Fig. 1a, b). Two other long α-helices (TM3 and TM5) also extend beyond the membrane plane. Within the membrane, α-helix interactions are mediated by both hydrophobic and hydrophilic contacts. A remarkable feature of all three TTYH paralogs concerns the presence of two conserved acidic residues in the TMD, one located on the extracellular border and another in the center of the hydrophobic core of the membrane (Fig. 2c and Supplementary Fig. 1a). Both residues appear to stabilize the structure by interaction with close-by polar residues located on neighboring transmembrane helices (Fig. 3a). Apart from these polar islets, the remainder of the helix bundle is predominantly hydrophobic. In the center of the lipid bilayer, the four helices are tightly packed, whereas the packing loosens on the intracellular side, where TM2 and TM3 have detached, and at the extracellular entrance, where TM3 and TM4 have splayed apart to open a gap toward the membrane that seamlessly leads into a spacious cavity in the extracellular domain (Figs. 2c, d and 3a). This large cavity, which measures about 36 Å in the long direction protrudes from the membrane and is contained within the subdomain Ex1. It is on one side lined by helices TM2/E1, TM3, and TM5 and on the other side covered by an extended region that contains a helical hairpin of the short helices E3 and E4 (Fig. 2d). The latter resembles a flap, which is stabilized on both sides by conserved disulfide bridges confining its observed position in the structure (Fig. 2d). In none of our datasets, we have observed a pronounced mobility of this region (Supplementary Figs. 2–6). The described cavity contains polar and charged residues on its membrane boundary, whereas it is lined by hydrophobic residues for the remainder of the structure (Supplementary Fig. 7c). Together, these structural features bear characteristics of an interaction site with unknown ligands or substrates. Finally, the distal extracellular region Ex2 is organized as tightly interacting four-helix bundle formed by the connected α-helix pairs E1/E2 and E5/E6 (Fig. 2a, e). On their part located closer to the membrane, E5 and E6 mediate subunit interactions, whereas they diverge toward the periphery of the protein thus providing the characteristic Y-shape of the molecule that is apparent in views along the long dimension of the dimeric protein (Fig. 1b). The bulk of the dimer interactions in the extracellular domain are formed by charged and polar residues (Supplementary Fig. 7d–f). Although the largest fraction is

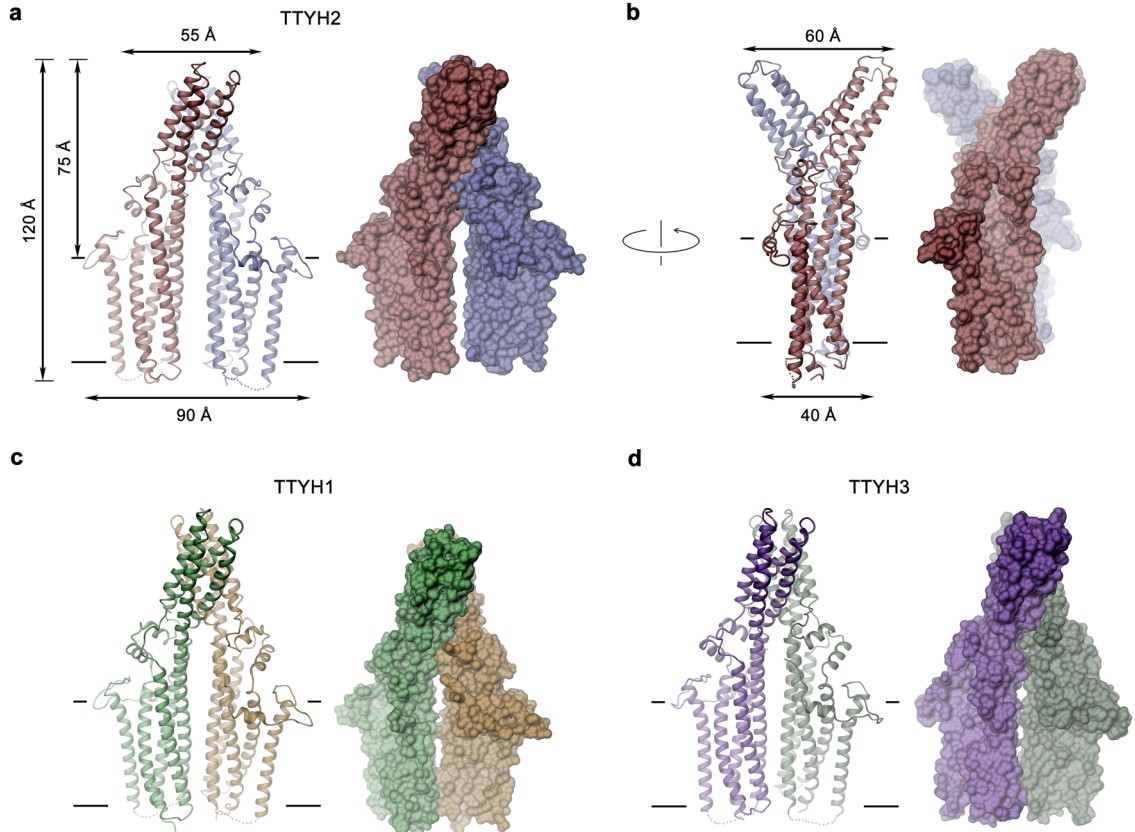

**Fig. 1 Structure of the TTYH dimer. a–d** Ribbon representation (left) and molecular surface (right) of TTYH1 (**a**, **b**), TTYH2 (**c**), and TTYH3 (**d**). The view is from within the membrane, in **a**, **c**, and **d** along the short and in **b** along the long dimension of the protein. **a**, **b** Molecular dimensions are displayed. **a–d** Membrane boundaries are indicated as lines. Subunits are shown in unique colors.

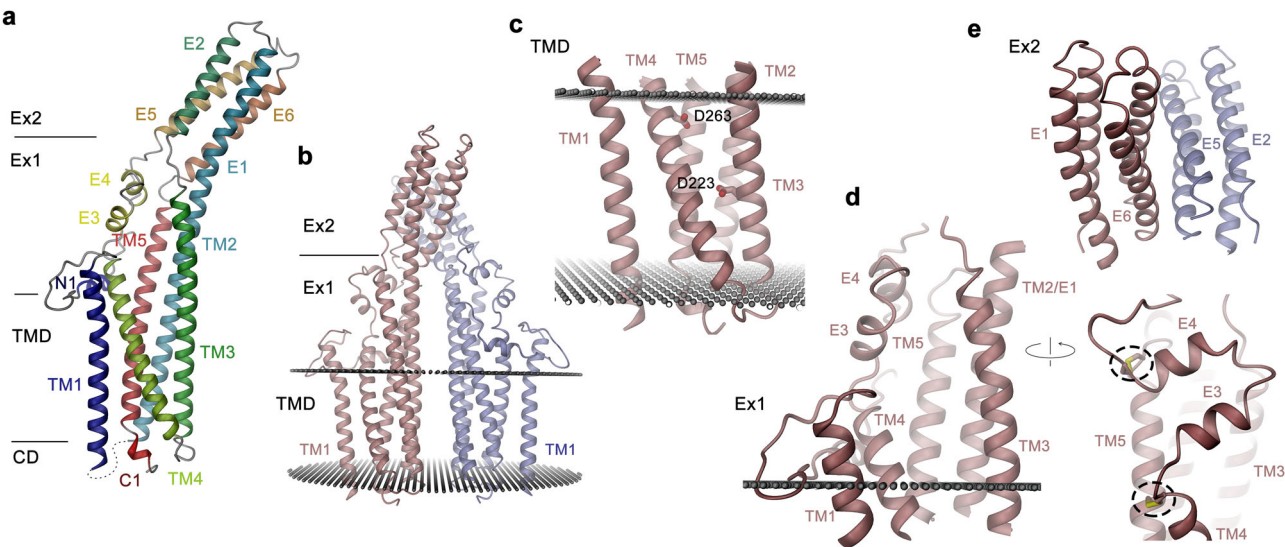

**Fig. 2 Subunit structure. a** Ribbon representation of the TTYH2 subunit with secondary structure elements labeled and shown in unique colors. **b** Structure of the TTYH2 dimer. **c–e** Structure of different domains of TTYH2. **c** Transmembrane domain (TMD) with acidic residues displayed as sticks and labeled, **d** structure of the extracellular proximal subdomain. The view is toward the hydrophobic cavity. Inset (right) shows a blow-up of the one border of the cavity lined by the E3-E4 hairpin, which is stabilized by disulfide bridges (highlighted by dashed ovals). **b–d** Boundaries of the hydrophobic membrane core are shown as spheres. **e** Dimeric structure of the distal extracellular subdomain Ex2, which constitutes the dimer interface.

contributed by residues of Ex2 located on α-helices E5 and E6 (Supplementary Fig. 7e), single contacts are also found in Ex1 (Supplementary Fig. 7f). Toward the membrane, an excess of acidic residues facing the dimer interface renders the electrostatics strongly negative (Supplementary Fig. 7g–j). As observed in the data from TTYH3, these residues potentially coordinate $Ca^{2+}$ ions, which might stabilize the observed dimeric structure (Supplementary Fig. 7g). In contrast to contacts in the

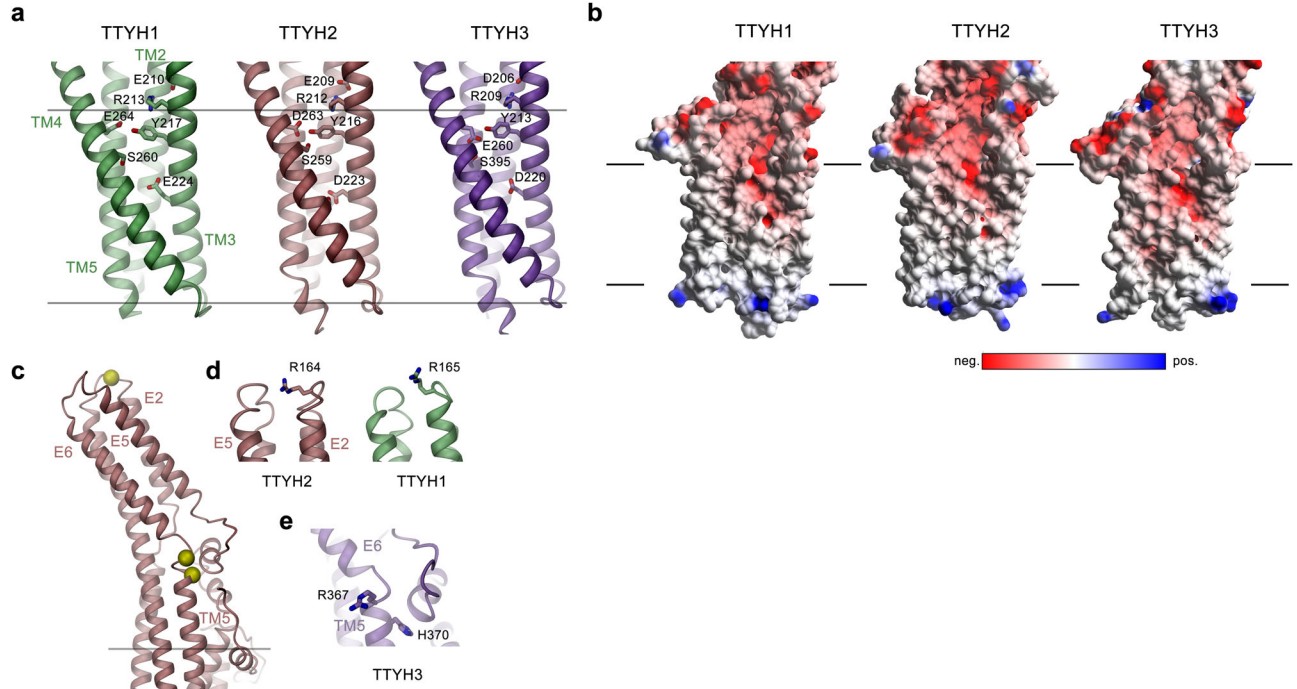

**Fig. 3 Features of the TM domain. a** Interactions of the membrane-spanning segments TM2–5 in the three TTYH paralogs. Sections of the protein are displayed as ribbon with sidechains of polar and charged residues shown as sticks and labeled. **b** Section of the TM region of TTYH paralogs. The protein is displayed as surface, with the electrostatic potential mapped (red negative, blue positive). The view is as in **a**. **c** Extracellular domain of TTYH1 with residues that were predicted to line the pore displayed as yellow spheres. **d**, **e** Blow-up of proposed pore regions of TTYH2 and TTYH1 (**d**) and TTYH3 (**e**). Predicted pore lining residues are shown as sticks. **a**–**c** The boundaries of the hydrophobic membrane core are indicated by lines.

extracellular domain, the interactions between subunits within the membrane region observed for TTYH1 and 3 are hydrophobic (Supplementary Fig. 7k). The extracellular regions Ex1 and Ex2 and the N-terminus preceding TM1 also contain conserved glycosylation sites that are clearly manifested in the cryo-EM density (Supplementary Figs. 2f and 5f).

**Investigation of ion channel function.** The structural features of all three proteins are puzzling in light of their proposed role as chloride channels[3,18]. Whereas several families of anion transport proteins share a similar dimeric organization[21–24], with independent permeation paths being contained within individual subunits[25–27], the respective TMDs of TTYHs do not exhibit pronounced characteristics of an anion conduction pore, which would presumably be located between the interacting helices TM2–TM5 (Fig. 3a). In the center of the membrane, this region is tightly packed and would have to undergo a major rearrangement to open an aqueous pore. Moreover, due to the excess of acidic residues at the extracellular side close to the membrane and within the TMD, the electrostatic environment is strongly negative, which would further repel permeating anions (Fig. 3a, b and Supplementary Fig. 7h–j). Finally, the residues proposed in previous studies to line an ion conduction path[3,18] are remote from the membrane and would not likely exert a pronounced influence on potential transport processes (Fig. 3c). The respective residues on TTYH1 or TTYH2 (ref. [18]) are located on Ex2 at the loop connecting E1 and E2 at the extracellular periphery of the protein (Fig. 3d) and ones suggested for TTYH3 (ref. [3]) on Ex1 close to the dimer interface (Fig. 3e).

In light of the unexpected architecture of the TTYH family, we revisited the experiments that have led to the proposal that the proteins form ion channels. $Ca^{2+}$-activated chloride currents attributed to either TTYH2 or 3 were initially recorded by patch-clamp electrophysiology from CHO cells transfected with constructs coding for the respective proteins[3,17]. Ion conduction was observed in the whole-cell configuration and in inside-out patches in the presence of 0.5 mM $Ca^{2+}$ on the intracellular side. In a later study, swelling-activated channel activity of all three paralogs was observed in astrocytes, but the same functional behavior could be transferred to HEK293 cells transiently expressing TTYH proteins, where endogenous levels of LRRC8 proteins were downregulated by co-transfection of corresponding short hairpin RNAs[18]. Opposed to LRRC8 channels, these putative TTYH-related currents were observed in response to severe cell swelling but not upon reduction of the intracellular ionic strength, which is sufficient for LRRC8 activation[28,29]. We thus initially attempted to reproduce the swelling-induced activation of anion channels in wild-type HEK293 cells by patch-clamp electrophysiology in the whole-cell configuration and found the emergence of large currents, which appeared after a 60 s incubation period following the exposure of cells to hypotonic conditions (HPC, Fig. 4a). These currents are mediated by LRRC8 channels, which is manifested in their absence in a HEK293 cell line where all five LRRC8 genes were knocked out (LRRC8$^{−/−}$)[30] and their re-emergence in the same cell line upon transfection with DNA coding for the paralogs LRRC8A and C (Fig. 4a, b)[28,30]. As in non-transfected LRRC8$^{−/−}$ cells, we did not record any response in the same cells expressing one of the three TTYH paralogs even after a prolonged exposure to HPC (Fig. 4c–f). This lack of activity is irrespective of visible cell swelling and the robust expression of all three proteins at the plasma membrane, which was confirmed for TTYH2 and TTYH3 in surface biotinylation experiments (Fig. 4d–e). We also attempted to measure the described swelling-activated currents upon co-expression of the aquaporin (aquaporin 4 (AQP4)), which further increases the water permeability of HEK293 cells[18]. Also in this case, we did not find any compelling evidence of swelling-induced currents that are mediated by TTYH proteins

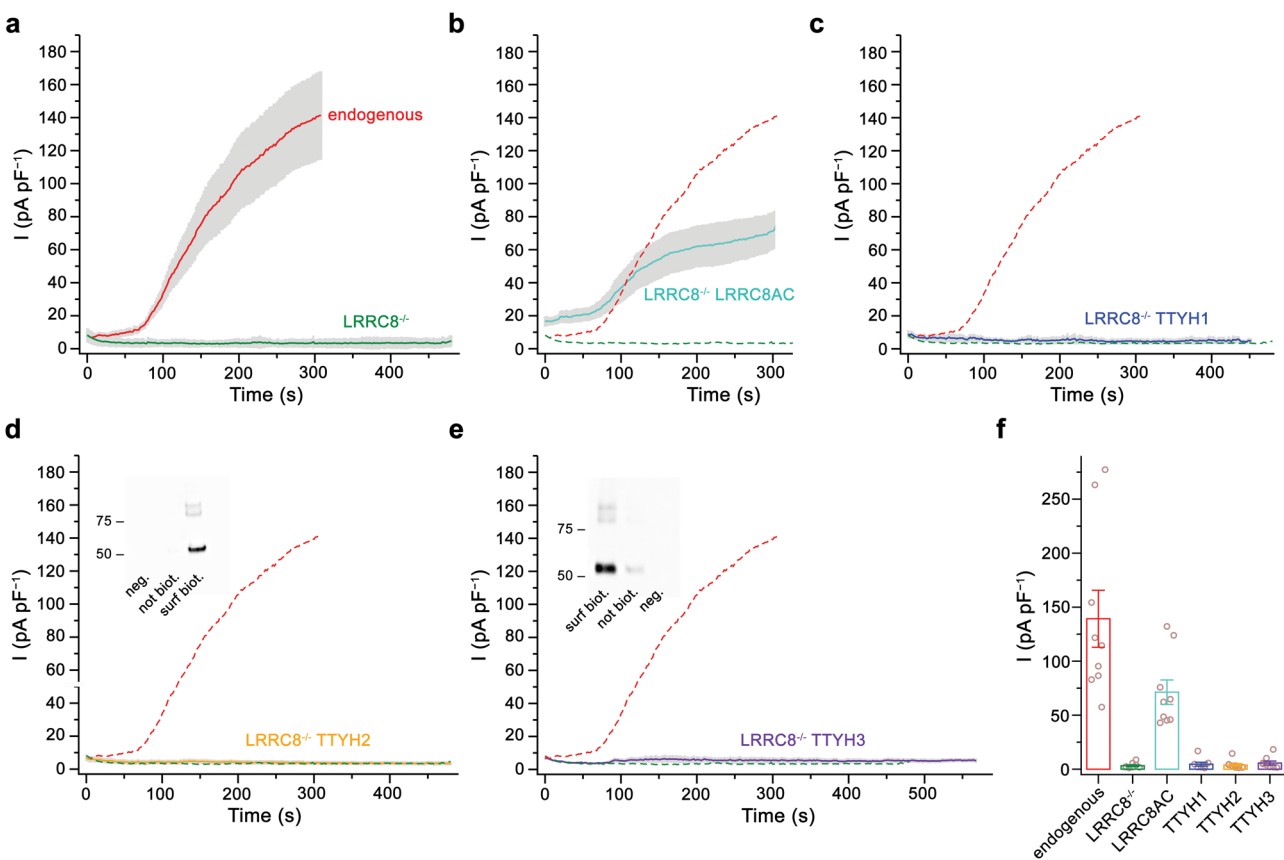

**Fig. 4 Investigation of ion channel function.** Current response of cells exposed to hypotonic conditions (starting at $t = 0$ and lasting for the entire duration of the recording). Currents are measured by patch-clamp in the whole-cell configuration at 100 mV. **a–e** Average current density of **a** HEK293 cells (endogenous, $n = 9$) and LRRC8$^{-/-}$ cells ($n = 9$), LRRC8$^{-/-}$ cells transfected with **b** LRRC8A and LRRC8C ($n = 9$), **c** TTYH1 ($n = 8$), **d** TTYH2 ($n = 10$), and **e** TTYH3 ($n = 9$). **b–e** Mean values of endogenous currents of HEK293 cells (red) and LRRC8$^{-/-}$ cells (green) shown in **a** are displayed as dashed line for comparison. **d**, **e** Inset shows western blots from a surface expression analysis of the respective constructs detected by biotinylation. Approximate molecular weights (kDa) of marker proteins are indicated. **f** Current response (at 100 mV) of cells expressing the indicated constructs recorded 300 s after exposure to hypotonic medium (endogenous, $n = 9$; LRRC8$^{-/-}$, $n = 9$; LRRC8AC, $n = 9$; TTYH1, $n = 8$, TTYH2, $n = 10$; TTYH3, $n = 9$). Values from individual measurements are shown as circles, mean values as bars. Differences compared to LRRC8$^{-/-}$ cells were analyzed in a two-sample two-sided $t$ test and found to be non-significant for cells expressing TTYH constructs (**f**, endogenous $p = 0.0009$, LRRC8AC $p = 0.0003$, TTYH1 $p = 0.50$, TTYH2 $p = 0.67$, TTYH3 $p = 0.24$). **a–f** errors are s.e.m.

(Supplementary Fig. 8a–c). In related experiments, we were also not able to reproduce a previously described phenotype of TTYH-mediated chloride conduction that is activated by increased intracellular Ca$^{2+}$ (ref. [3]) by recording of whole-cell currents in TTYH-expressing LRRC8$^{-/-}$ cells with pipette solutions containing up to 1 mM of free Ca$^{2+}$ (Supplementary Fig. 8d–h).

Although the structures of TTYH proteins do not show characteristic features of anion channels, several properties, such as the negative electrostatic potential at the extracellular part of the TMD and the presence of conserved acidic residues within the hydrophobic membrane core could be compatible with the transmembrane flow of protons (Fig. 3a, b). We thus investigated whether TTYHs might facilitate H$^+$ transport. For that purpose, we have recorded H$^+$ currents in LRRC8$^{-/-}$ cells expressing TTYH1–3 by electrophysiology in the absence and presence of Ca$^{2+}$ using a protocol established for the characterization of voltage-gated proton channels[31] (Supplementary Fig. 9a–j) and after reconstitution of purified TTYH1–3 into liposomes by a fluorescence-based transport assay, which was previously used to monitor electrogenic H$^+$ transport in proton channels[32] and proton-coupled transporters[33] (Supplementary Fig. 9k–m). In neither case did we find evidence for transmembrane H$^+$ transport. In light of its distinct protein architecture and the

absence of functional evidence for ion transport, we thus conclude that the TTYH family does not form ion channels and instead confers other catalytic properties.

**The extracellular cavity as binding site of hydrophobic substances.** An intriguing common feature of TTYH structures concerns the large and solvent-exposed cavity located on the extracellular side, which, irrespective of local differences between TTYH1, 2, and 3, shares close resemblance between the three paralogs (Fig. 5 and Supplementary Fig. 10). This cavity extends from the outer leaflet of the bilayer but is for its largest part contained within the proximal extracellular domain Ex1 (Fig. 2d). Since its membrane-embedded inner boundary is delimited by charged and hydrophilic residues reminiscent of a lipid scramblase[34] (Fig. 3a), we were interested whether TTYH proteins would facilitate the movement of lipids between both leaflets of the bilayer. We have investigated this question by employing an assay that was previously used to characterize the functional properties of lipid scramblases of the TMEM16 family[34–36]. For that purpose, we have reconstituted TTYH2 into liposomes containing traces of fluorescently labeled lipids and assayed the bleaching of lipids residing in the outer leaflet in the absence and presence of Ca$^{2+}$. However, in no case did we find any evidence

for TTYH2 to facilitate lipid flip-flop (Supplementary Fig. 11). We thus continued to investigate the part of the described cavity that is located outside of the membrane and exposed to the surrounding aqueous environment. At its extracellular end, this extended pocket is confined by the E2–E3 flap and helices TM2/E1 and TM3 (Fig. 2d). Its inside is predominantly hydrophobic, whereas the rims are surrounded by hydrophilic amino acids (Fig. 5a and Supplementary Fig. 7c). Despite the generally apolar character, the long-range electrostatics conferred by conserved close-by negatively charged residues, residing on TM2 and TM3 and facing the dimer interface, might extend into the cavity and contribute to a negative potential in its interior (Fig. 3b and Supplementary Fig. 7h–j). Owing to their physico-chemical properties, the respective pockets in all three proteins appear to be suited to interact with hydrophobic compounds that might also reside in the lipid bilayer. In their shape and with respect to general structural properties, they resemble sterol-binding sites observed in transport proteins[37] and they would be of appropriate size to harbor structurally related compounds (Supplementary Fig. 10a). In cryo-EM maps of the three paralogs obtained in detergent, the pocket is filled with residual density reflecting its occupancy by small molecules. In TTYH2, six elongated density fragments line the entire pocket perpendicular to its long dimension (Fig. 5b and Supplementary Fig. 10e). This distribution is largely preserved in the dataset of TTYH2 that was reconstituted into lipid nanodiscs assembled from a phospholipid–cholesterol mixture (Supplementary Fig. 10f). Finally, we find a similar distribution of residual density also in the data of TTYH3, thus underlining that this feature reflects a property that is shared between paralogs (Fig. 5c and Supplementary Fig. 10g). Although the insufficient resolution prohibits the assignment of their chemical properties with confidence, some of the densities could be of appropriate size of a sterol ring system, whereas other fragments resemble less bulky extended aliphatic chains, which point toward lipid-like structures (Fig. 5b, c and Supplementary Fig. 10e–g). While residual density is also found in the data of TTYH1, its detailed distribution differs owing to the distinct shape of the cavity at its extracellular end (Fig. 5d and Supplementary Fig. 10h). As all proteins were purified in the detergent GDN, we cannot exclude that the observed density would reflect the interaction with its cholesterol-derived diosgenin moiety or breakdown products thereof. Still, the presence of very similar densities even after reconstitution of TTYH2 into lipid nanodiscs and removal of excess detergent by biobeads indicates the intrinsic affinity of the region to interact with small molecular lipid-like compounds (Supplementary Fig. 10f). To investigate the apolar content of molecules that are co-purified

with TTYH2, we have prepared methanol and chloroform extracts of a purified sample and analyzed its composition by liquid chromatography and mass spectrometry (LC-MS). This analysis revealed the presence of molecules that are enriched in the protein extract compared to the background of the GDN buffer. The most abundant compounds that are not found in the buffer could be assigned to phospholipids, sphingolipids, and prostaglandins (Supplementary Fig. 12) thus further emphasizing the potential role of TTYH molecules to provide interaction platforms with lipids and lipid-like molecules.

## Discussion

The present study was motivated by our interest in the TTYH family, whose members are highly expressed in the brain[5–8]. Despite their abundance, little was known about the molecular features of TTYHs as, with respect to sequence, they are unrelated to other proteins. Although proposed to form chloride channels that are activated by either $Ca^{2+}$ or cell swelling[3,18], the detailed functional properties and their relationship to structure has remained elusive.

To reveal their unknown architecture, we have determined cryo-EM structures of all three paralogs (Supplementary Figs. 2–6). These structures depict the common molecular features of a conserved protein family. All TTYH molecules are dimers and contain an extended extracellular domain, five membrane spanning segments, and an unstructured cytoplasmic region (Figs. 1 and 2). In all cases, the predominant part of the dimer interaction is formed on the extracellular side. Additionally, there are contacts between the TMDs of TTYH1 and 3 but not of TTYH2 (Fig. 1 and Supplementary Fig. 7h–j). Since we do not detect pronounced structural differences upon incorporation of TTYH2 in a lipid environment or addition of $Ca^{2+}$ to the sample of TTYH3, we are confident that the observed structures provide a representative view of the entire family (Fig. 1 and Supplementary Figs. 2–6).

An evident feature of all structures was their lack of resemblance to an anion-conducting channel (Figs. 1 and 2). The dimeric organization of the protein contradicts the previously proposed higher oligomeric state of subunits that assemble like the staves of a barrel to enclose a single ion conduction pore running along their symmetry axis[18] as observed in hexameric channels of the LRRC8 family[28] or pentameric Bestrophins[38] and GABA receptors[39]. However, this alone would not preclude ion channel function since several chloride channel and transport proteins are dimers, such as members of the CLC, TMEM16, SLC12, and SLC26 families, where the ion transport unit is usually contained within a single subunit[21–24]. In case of the

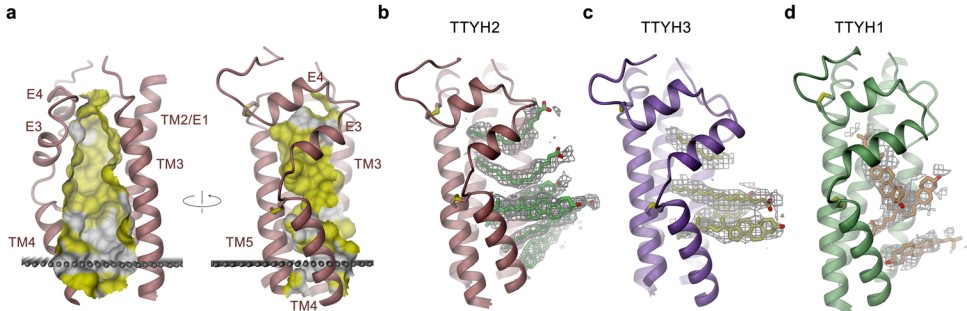

**Fig. 5 Structural features of the extracellular cavity. a** Views of the extracellular cavity with the indicated relationship. The protein is represented as ribbon. The molecular surface of the cavity is shown. Regions contacted by hydrophobic and aromatic residues are colored in yellow. Gray spheres indicate the outer boundary of the hydrophobic membrane core. **b–d** Distribution of residual density in the extracellular cavity of TTYH2 (**b**), TTYH3 (**c**), and TTYH1 (**d**). The view of the protein is as in **a** (right). Cryo-EM density contoured at 5.5$\sigma$ is shown as gray mesh. Depending on the volume of the density, it was arbitrarily interpreted as either cholesterol or an acyl chain for size comparison.

TTYHs, several structural features of the subunit are inconsistent with the requirements for such an anion conduction pathway. With five TM helices, the membrane-inserted part of the protein is considerably smaller than that of other chloride transport proteins and it does not show any features reminiscent of an aqueous conduit (Fig. 3a). Moreover, the strong negative electrostatic potential at both entrances to the membrane region and within the membrane, which is conferred by an excess of acidic residues placed on the extracellular side close to the membrane boundary, would further impede anion conduction by electrostatic repulsion[40–42] (Fig. 3b and Supplementary Fig. 7h–k). Finally, none of the positions predicted to line the pore[3,18] are anywhere close to the membrane in positions where they could influence permeation (Fig. 3c). We thus revisited the proposed chloride channel function making use of a cell line where the five LRRC8 subunits forming volume-regulated chloride channels have been knocked out[30]. This cell line proved essential for the investigation of TTYH activity, since it completely abolished the predominant VRAC currents that are evoked upon cell swelling. In this system, we did not find any activity inferred by TTYH proteins despite their robust surface expression (Fig. 4 and Supplementary Fig. 8). While TTYH proteins do not show expected features of anion channels, the presence of conserved acidic residues residing in the membrane share some resemblance to a proton conduction pathway. Although our experiments did not provide any evidence of constitutive or $Ca^{2+}$-activated proton transport (Supplementary Fig. 9), they do not exclude the possibility of a channel gated by a different ligand or the coupling of $H^+$ flux to another substrate. Similar to the absence of ion conduction, we did not find any evidence of TTYH proteins mediating lipid flip-flop between both leaflets of the bilayer (Supplementary Fig. 11).

In light of the missing evidence for ion conduction or lipid scrambling, the functional role of TTYH proteins remains elusive. One of the most striking features of TTYHs concerns the presence of a conserved hydrophobic pocket that emerges from the membrane and leads to the extracellular domain (Figs. 2d and 5a). This pocket bears features of a site that interacts with substrates in a broader sense. Although of appropriate size and geometry to accommodate cholesterol or a chemically related compound (Supplementary Fig. 10a–d), we did not find evidence of tight sterol interactions, as these are not among the predominant fraction of molecules co-purified with the protein identified by mass spectrometry (Supplementary Fig. 12). Instead, we find residual density filling the extended pocket (Fig. 5b–d and Supplementary Fig. 10e–h) hinting at the presence of interacting amphiphilic molecules that are oriented perpendicular to the membrane potentially exposing their hydrophilic groups into the aqueous solution. Although intriguing, it is currently unclear whether this density reflects proper substrate interactions of the site or whether it corresponds to low-affinity binding of compounds in the absence of a specific ligand.

For the functional role of TTYHs, we thus envision three potential scenarios of a protein involved in transport, signaling, or catalysis of membrane-embedded compounds (Fig. 6). A potential role in lipid transport is supported by the distribution of densities in the extended pocket, which can be attributed to potential substrates. This pocket is reminiscent of a delocalized binding site for hydrophobic and amphiphilic molecules that might also reside in the membrane (Fig. 5). Since it emerges from the outer leaflet of the membrane, it might offer a diffusive pathway for the extraction of molecules from the lipid bilayer to potentially relay them to other interaction partners. In this respect, TTYHs resemble a lipid scramblase, although with inverted transport properties. Whereas a scramblase provides a membrane-embedded hydrophilic path for polar lipid headgroups[34], which facilitates their shuttling between the two leaflets of the bilayer, the hydrophobic cavity of TTYHs might offer a similar low-energy path for the hydrophobic portion of membrane-embedded compounds to diffuse toward an aqueous environment (Fig. 6a). Alternatively, instead of functioning as a delocalized binding region, the site might offer specific and tight interactions with an unknown substrate. In this respect, TTYHs could act as receptors that relay a signal upon ligand binding to still unidentified interaction partners that could either interact with their extracellular unit or the disordered cytoplasmic C-terminus (Fig. 6b). Finally, the proteins could act as enzymes that are involved in the turnover of a substrate (Fig. 6c) with polar residues inside the TMD and at the inner border of the pocket constituting an active site required for chemical catalysis (Fig. 3a).

Collectively, our study has uncovered the unknown architecture of TTYH proteins. Although our results refute a previous claim that these proteins function as pore-forming subunits of ion channels[3,18], we do not want to exclude the possibility that they might associate with such channels to act as accessory subunits involved in their regulation. Based on their structural features, we instead propose that TTYH proteins would be involved in lipid transport, signaling, or catalysis. To clarify the ambiguity concerning their function, the subcellular localization of the protein and the identification of interaction partners will be an important subject of future investigations, which should provide further evidence for the physiological role of these proteins. Our data provide a solid basis for such studies on an abundant but still poorly understood membrane protein family.

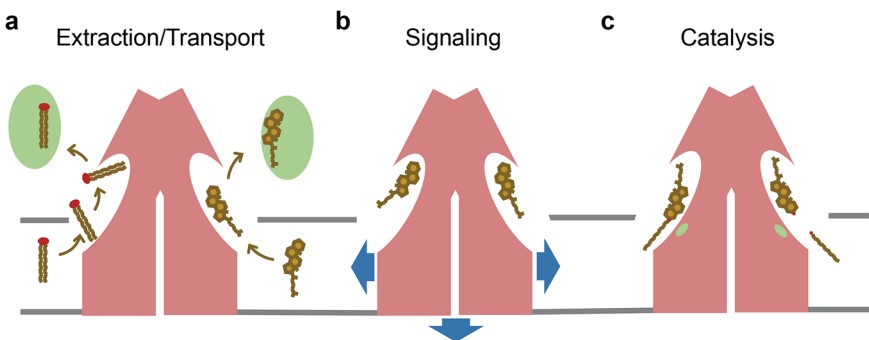

**Fig. 6 Potential functions of TTYH proteins. a** Based on the structural features of the extracellular cavity, TTYH proteins could be involved in the extraction of hydrophobic compounds from the membrane and their potential transfer to acceptor molecules. **b** Alternatively, TTYH proteins could act as receptors for unknown ligands and thus be involved in the transduction of signals into the cell. **c** Polar residues on the inside of the cavity close to the membrane boundary could act as an active site of an enzyme modifying, membrane-associated molecules.

## Methods

**Expression constructs.** All constructs were generated using FX-cloning and FX-compatible vectors[43]. Genes encoding full-length human TTYH proteins and the human AQP4 were codon-optimized for expression in human cell lines using the codon optimization tool from Integrated DNA Technologies and synthesized by GenScript. The genes were cloned into a pcDX vector containing a C-terminal Rhinovirus 3C Protease linker followed by Venus[44] in case of TTYH genes and mCherry in case of AQP4 (ref. [45]), a Myc-tag, and a streptavidin-binding peptide (SBP)[46]. For electrophysiological recordings, surface expression analysis, and liposome reconstitution, the TTYH genes were cloned into an analogous vector not containing Venus. Genes encoding murine LRRC8A and C cloned into the same vector were used for electrophysiological experiments[28]. The Venus-only construct used in electrophysiology contained the Venus gene followed by a Myc-SBP tag.

**Expression and purification of TTYH proteins and nanodisc reconstitution of TTYH2.** Suspension HEK293 GnTI⁻ cultures were grown in HyCell TransFx-H (Cytiva) media supplemented with 1% fetal bovine serum (FBS), 4 mM L-gluta-mine, 0.4% Poloxamer 188, and 100 U ml⁻¹ penicillin–streptomycin at 37 °C and 5% $CO_2$. For protein expression, cells were transiently transfected using PEI MAX (Chemie Brunschwig AG) and plasmid DNA purified from *Escherichia coli* MC1061 using the NucleoBond Gigaprep Kit (Machery-Nagel). Cells were harvested 60 h post-transfection, resuspended in lysis buffer (50 mM HEPES pH7.5, 200 mM NaCl, 2 mM EDTA, 1% GDN) supplemented with a cocktail of protease inhibitors, and solubilized for 3 h while rotating. The lysate was centrifuged at 20,000 × g for 30 min. The solubilized TTYH proteins were captured on Streptactin superflow resin (IBA Lifesciences) by batch-binding and washed with SEC buffer (10 mM HEPES pH 7.5, 200 mM NaCl, 2 mM EDTA, 50 μM GDN). Bound proteins were eluted with 5 column volumes of the SEC buffer supplemented with 5 mM D-desthiobiotin and digested with 3C protease at a TTYH:3C molar ratio of 1:2 for 1 h on ice. Eluted proteins were concentrated (Amicon, 100 kDa molecular weight cut-off), purified by SEC on a Superose 10/300 column (Cytiva), concentrated and used for the preparation of cryo-EM grids.

For nanodisc reconstitution, SEC-purified TTYH2 was mixed with MSP1D1 and a lipid mixture containing a 3:1:1:0.5 weight ratio of 1-palmitoil-2-oleoyl-sn-glycero-3-phosphoethanolamine (POPE, Avanti), 1-palmitoyl-2-oleoyl-sn-glycero-3-phospho-(1′-rac-glycerol) (POPG, Avanti), L-α-phosphatidylcholine (eggPC, Avanti), and cholesterol. MSP1D1 was expressed and purified as described[47], without cleaving the polyhistidine-tag. Briefly, the protein containing a His₆-tag on its C-terminus was expressed in *E. coli* and purified by affinity chromatography on Ni-NTA resin in buffer containing 20 mM HEPES pH7.5, 100 mM NaCl, and 0.5 mM EDTA. For reconstitution, an MSP:lipid:TTYH2 molar ratio of 3:433:1 was used. After mixing TTYH2 and lipids, the reconstitution sample was rotated for 30 min at room temperature (RT), then MSP1D1 was added and the sample was further rotated for 30 min. After addition of 0.2 mg ml⁻¹ of biobeads, the sample was transferred to 4 °C and incubated overnight. For cryo-EM sample preparation, TTYH2-containing nanodiscs were separated from empty nanodiscs by loading the reconstitution sample on Streptactin superflow resin and washing it extensively. TTYH2-containing nanodiscs were eluted with the SEC buffer supplemented with 5 mM D-desthiobiotin and concentrated to 1.5 mg ml⁻¹ for cryo-EM sample preparation.

**Cryo-EM grid preparation and data acquisition.** For structure determination of TTYH1, 2 and 3 in GDN and TTYH2 in lipid nanodiscs by cryo-EM, 2.5 μl samples at 1.5–2.1 mg ml⁻¹ were applied on glow-discharged holey carbon grids (Quantifoil R1.2/1.3 or R0.6/1 Au 200 mesh) and the sample excess was removed by blotting for 3–7 s in a controlled environment (4 °C and 100% relative humidity). Grids were flash-frozen in liquid ethane/propane mixture using Vitrobot Mark IV (Thermo Fisher Scientific). Cryo-EM grids were imaged on a 300 kV Titan Krios G3i (Thermo Fisher Scientific) with a 100 μm objective aperture. All data were acquired using a post-column quantum energy filter (Gatan) with a 20 eV slit and a K3 summit direct detector (Gatan) in super-resolution mode. In case of TTYH2, TTYH1, and TTYH2 in nanodiscs datasets, dose-fractionated micrographs were recorded with a defocus range of –0.8 to –2.4 μm in an automated mode using EPU 2.7 + AFIS faster acquisition (Thermo Fisher Scientific). The TTYH3 dataset was recorded with a defocus range of −1 to −2.4 μm in an automated mode using EPU 2.9 (Thermo Fisher Scientific). All datasets were recorded at a nominal magnification of ×130,000 corresponding to a pixel size of 0.651 Å pixel⁻¹ (0.3255 Å pixel⁻¹ in super-resolution) with a total exposure time of 1.01 s (36 individual frames) and a dose of 1.69 e⁻ Å⁻² frame⁻¹ in case of TTYH2, TTYH1, and TTYH2 in nanodiscs datasets and a dose of 1.932 e⁻ Å⁻² frame⁻¹ in case of TTYH3. The total electron dose on the specimen level for all datasets except TTYH3 was approximately 61 e⁻ Å⁻² for TTYH3 dataset and the total electron dose was approximately 67 e⁻ Å⁻².

**Cryo-EM data processing.** Datasets of TTYH2 in GDN or nanodiscs were processed in RELION 3.0.9 (ref. [48]). TTYH1 and TTYH3 datasets were processed in RELION 3.1 (ref. [49]). All micrographs were preprocessed in the same manner using RELION's own implementation of MotionCor2 (ref. [50]) and CTFFIND-4.1 (ref. [51]).

Low-quality micrographs were manually removed from the TTYH2 in GDN dataset. In the other datasets, all micrographs were kept for further processing.

Particles from the dataset of TTYH2 in GDN were initially manually picked and two-dimensionally (2D) classified to generate templates for autopicking. Autopicking resulted in 3,500,000 particles, which were extracted with a box size of 440 pixels (286 Å) with 4× binning (110-pixel box size, 2.604 Å pixel⁻¹) and subjected to two subsequent rounds of 2D classification, where 644,000 selected particles were used to generate an initial model. The initial model served as a reference in a three-dimensional (3D) classification, where the best-looking class containing 267,000 particles was selected, the particles were unbinned 2 times (220-pixel box size, 1.302 Å pixel⁻¹) and refined to 3.3 Å with C2 symmetry imposed. The resulting TTYH2 map was used as a 3D reference for autopicking of particles in the other datasets.

For the datasets of TTYH2 in nanodiscs, TTYH1 and TTYH3 the same processing pipeline was used as for TTYH2 in GDN. Data processing details of each dataset are included in Supplementary Figs. 2–5. The TTYH1 dataset was a merge of two individually acquired and processed datasets. Processing of the individual TTYH1 datasets included extra 3D classification steps without image alignment for elimination of poorly aligned particles, which resulted in only 5% of the initial particles included in the refinement of the final TTYH1 map.

**Model building and refinement.** The TTYH2 model was built de novo into the cryo-EM density of TTYH2 in GDN at 3.3 Å in Coot[52]. The four glycosyl moieties visible in the cryo-EM density of TTYH2 served as starting points for placing asparagine residues predicted to be glycosylated. It was possible to unambiguously assign residues 6–72 and 89–416.

The TTYH2 model was used as a template for creating homology models of TTYH1 and TTYH3 using the Swiss-model webserver[53]. The generated homology models of TTYH1 and TTYH3 were fitted in the respective cryo-EM densities using rigid-body fit in UCSF Chimera[54] and Coot. The TTYH2 model was fitted into the cryo-EM density of TTYH2 in nanodiscs in the same manner. In the TTYH1 model, it was possible to unambiguously assign residues 5–70 and 90–422 and in the TTYH3 model—residues 4–70 and 87–414. All atomic models were refined in PHENIX[55] maintaining NCS and secondary structure restraints throughout (Table 1 and Supplementary Fig. 6). Membrane boundaries were estimated with the PPM server[56]. Figures and containing molecular structures and densities were prepared with DINO (http://www.dino3d.org), Chimera[54], and ChimeraX[57]. Surfaces were generated with MSMS[58]; electrostatic potentials were calculated and displayed in Coot.

**Surface expression analysis.** For surface expression analysis of TTYH proteins, HEK293 LRRC8-knockout cells (LRRC8⁻/⁻, kindly provided by T. J. Jentsch) were cultured in Dulbecco's Modified Eagle Medium (DMEM; Gibco) supplemented with 10% FBS and 100 U ml⁻¹ penicillin–streptomycin at 37 °C and 5% $CO_2$. The cells were transfected at 70% confluency 24 h prior to the analysis with 10 μg of individual TTYH constructs per dish using lipofectamine 2000 (Invitrogen). Proteins expressed on the cell surface were biotinylated with sulfo-NHS-SS-biotin using the Pierce Cell Surface Protein Isolation Kit (Thermo Fisher) according to the manufacturer's protocol. For the non-biotinylated control, the cells were treated with phosphate-buffered saline instead of biotin. The cell pellet from one 10-cm dish was resuspended in 0.4 ml lysis buffer (50 mM HEPES pH7.5, 200 mM NaCl, 2 mM EDTA, 1% GDN) and incubated while rotating at 4 °C for 1 h. Extracted protein was harvested by centrifugation at 10,000 × g for 10 min and 0.2 ml of the supernatant was used for binding to 50 μl bed volume of the SEC buffer-equilibrated NeutrAvidin resin for 1 h at 4 °C under constant rotation. The flow-through was discarded and the resin was washed 5 times with the SEC buffer. Bound protein was eluted by incubation with SEC buffer supplemented with 50 mM dithiothreitol for 1 h at 4 °C under constant rotation. The samples were separated by sodium dodecyl sulfate–polyacrylamide gel electrophoresis (SDS-PAGE), transferred to a polyvinylidene fluoride membrane (Merck Millipore), and analyzed by western blotting using a mouse monoclonal Anti-c-Myc primary antibody at a 1:5000 dilution (Sigma, M4439, clone 9E10) and a goat anti-mouse-horseradish peroxidase secondary antibody at a 1:10,000 dilution (Jackson ImmunoResearch). The chemiluminescent signal was developed with Western-nBright™ Quantum substrate (Advansta) and imaged with a Fusion FX7 system (Vilber).

**Proton flux assay.** TTYH1, 2, or 3 was reconstituted into liposomes for H⁺ flux assays. Empty liposomes were prepared by mixing POPE (Avanti), POPG (Avanti), eggPC (Avanti), and cholesterol in 3:1:1:0.5 weight ratio and evaporating chloroform with a stream of nitrogen gas followed by overnight desiccation. In case of TTYH2 and 3, the liposomes did not contain cholesterol. The resulting lipid film was resuspended in 20 mM HEPES, pH 7.0, 150 mM KCl, 0.2 mM EDTA (liposome buffer) to 20 mg ml⁻¹ by sonication. The liposomes were extruded through 200 nm polycarbonate membrane, flash-frozen in liquid nitrogen, and stored at −80 °C.

For protein reconstitution, empty liposomes were destabilized by addition of 10% TritonX-100 while monitoring the turbidity at 540 nm. TTYH1, 2, or 3 purified as described before were added to the destabilized liposomes at a 1:100 protein:lipid ratio and rotated for 15 min at RT. The first portion of biobeads (40

mg ml$^{-1}$) was added and the reconstitution was rotated for another 20 min at RT. Subsequently, the second portion of biobeads was added and the reconstitution was transferred to 4 °C. After 1 h of rotation, another portion of biobeads was added and the reconstitution was incubated overnight. The next day, a final portion of biobeads was added and after 2 h incubation liposomes were harvested by ultracentrifugation at 170,000 × g for 30 min at 23 °C and resuspended in liposome buffer to a concentration of 20 mg ml$^{-1}$. Presence of protein was confirmed by SDS-PAGE. Mock liposomes without protein were produced following the same procedures.

To probe the H$^+$ flux mediated by TTYH proteins, we used a liposome system with an asymmetric concentration of K$^+$ ions creating a membrane potential that favors the influx of protons, as described by Lee et al.[32]. Proton influx was detected with a membrane-permeable 9-amino-6-chloro-2-methoxyacridine (ACMA, Thermofisher) fluorophore. Upon protonation, the fluorescence of ACMA is quenched and the molecule becomes membrane-impermeable.

For transport experiments 20 µl of liposomes either containing or lacking protein were briefly sonicated until becoming translucent; diluted 20 times in 20 mM HEPES pH 7.0, 150 mM NaCl, 7.5 mM KCl, 0.2 mM EDTA, and 2 µM ACMA; and aliquoted into a black 96-well plate (Corning). The fluorescence was detected in 5 s intervals with excitation and emission wavelengths of 412 and 482 nm, respectively, on a Tecan Infinite M 1000 spectrofluorometer. After recording a baseline for 150 s, 20 nM of valinomycin was added to establish a membrne potential to initiate the H$^+$ flux. After recording the fluorescence for another 360 s, 2 µM of CCCP was added to dissipate the H$^+$ gradient. Proton-flux data are accessible via the Dryad database[59].

**Scrambling assay**. To test whether TTYH2 facilitates lipid scrambling, the protein was reconstituted into liposomes prepared from soy extract polar lipids (Avanti) doped with 0.5% w/w 1-oleoyl-2-(6-(7-nitro-2-1,3-benzoxadiazol-4-yl)aminohexanoyl)-sn-glycero-3-phosphoethanolamine (18:1-06:0 NBD-PE, Avanti). Empty liposomes and liposomes containing protein at a 1:100 protein:lipid mass ratio were prepared as described for the proton flux assay. The samples were protected from light to prevent bleaching of NBD groups. The scrambling assay was performed as described[34–36]. Briefly, liposomes were diluted in assay buffer (80 mM HEPES pH 7.5, 300 mM KCl, 2 mM EDTA, containing either 0 or 2 mM of free calcium) to a concentration of 0.2 mg ml$^{-1}$ and scrambling was monitored using excitation and emission wavelengths of 470 and 530 nm, respectively. After recording a baseline for 60 s, sodium dithionite was added to 30 mM, and the fluorescence was recorded for another 360 s. Scrambling data are accessible via the Dryad database[60].

**LC-MS analysis of lipids co-purified with TTYH2**. To identify lipid-like molecules co-purifying with TTYH2, we used chloroform and methanol extractions and analyzed the extracts with LC-MS. A total of 670 µg of TTYH2 purified as described before was divided into two parts for each extraction method. One part was extracted in 50% (v/v) chloroform. After phase separation, the organic phase containing lipids was collected and stored at −20 °C. The second part of the preparation was used for methanol extraction. In this case, 80% (v/v) methanol was added to the purified TTYH2, mixed, incubated for 10 min at RT, and stored at −20 °C. Prior to analysis, half of the total lipid extracts were dried under N$_2$ flow at 30 °C. Lipid samples were mixed with 100 µl of a 50% aqueous methanol solution at 20 °C while shaking at 600 rpm for 15 min. The samples were centrifuged at 20 °C for 10 min at 15,000 × g. For both extractions 75 µl of the supernatants were transferred to a glass vial with narrowed bottom (Total Recovery Vials, Waters) and subjected to LC-MS. Lipids were separated on a nanoAcquity UPLC (Waters) equipped with an HSS T3 capillary column (150 µm × 30 mm, 1.8 µm particle size, Waters), applying a gradient of 5 mM ammonium acetate in water/acetonitrile 95:5 (A) and 5 mM ammonium acetate in isopropanol/acetonitrile 90:10 (B) from 5% B to 100% B over 10 min. The following 5 min, conditions were kept at 100% B, followed by a 5 min re-equilibration to 5% B. The injection volume was 1 µl. The flow rate was constant at 2.5 µl min$^{-1}$. The UPLC was coupled to a QExactive mass spectrometer (Thermo) via a nano-ESI source. MS data were acquired using positive polarization and data-dependent acquisition. Full-scan MS spectra were acquired in profile mode from 106.7 to 1600 m/z with an automatic gain control target of 1 × 10$^6$, an Orbitrap resolution of 70,000, and a maximum injection time of 200 ms. The five most intensely charged ($z$ = +1 or +2) precursor ions from each full scan were selected for collision-induced dissociation fragmentation. Precursor ions were accumulated with an isolation window of 0.4 Da, an automatic gain control value of 5 × 10$^4$, a resolution of 17,500, and a maximum injection time of 50 ms and fragmented with a normalized collision energy of 10, 20, and 30 (arbitrary unit). Generated fragment ions were scanned in the linear trap. Minimal signal intensity for MS2 selection was set to 5 × 10$^3$. Lipid datasets were evaluated with the Progenesis QI software (Nonlinear Dynamics), which aligns the ion intensity maps based on a reference data set, followed by a peak picking on an aggregated ion intensity map. Detected ions were identified based on mass accuracy, detected adduct patterns, and isotope patterns by comparing with entries in the LipidMaps Data Base (LM). A mass accuracy tolerance of 5 mDa was set for the searches. Fragmentation patterns were considered for the identifications of metabolites. Matches were ranked based on mass error (observed mass − exact mass),

isotope similarity (observed versus theoretical), and relative differences between sample and blank. LC-MS data are accessible via the Dryad database[60].

**Electrophysiology**. We investigated the proposed chloride channel function of TTYH proteins with electrophysiological recordings of LRRC8$^{-/-}$ cells transfected with TTYH1, 2, or 3. The cells were cultured in DMEM (Gibco) supplemented with 10% FBS and 100 U ml$^{-1}$ penicillin–streptomycin at 37 °C and 5% CO$_2$. Approximately 24 h before recordings, cells were transfected using lipofectamine 2000 (Invitrogen) with 10 µg DNA of individual TTYH constructs per dish together with Venus-only construct to aid identification of transfected cells at a TTYH:Venus ratio of 3:1. For experiments with aquaporin, individual TTYH constructs were co-transfected with the AQP4-mCherry construct at a TTYH:AQP4 ratio of 1:1. As a positive control, we recorded currents in HEK293T cells endogenously expressing LRRC8 proteins.

Whole-cell patch-clamp recordings were performed on single cells with a seal resistance of ≥2 GΩ before establishment of the whole-cell configuration. Patch pipettes were pulled from borosilicate glass capillaries with an outer diameter of 1.5 mm and inner diameter of 0.86 mm (Sutter) and were fire-polished. Pipette resistance was typically 3–6 MΩ when filled with the intracellular solution. Series resistance was typically of 2–7 MΩ. Axopatch 200B and Digidata 1440 (Molecular Devices) were used for the recordings. Data were acquired using Clampex 10.7 (Molecular Devices). During recordings, cells were locally perfused using a gravity-fed system. For recordings of swelling-activated currents, the isotonic extracellular solution containing 10 mM HEPES-NMDG pH 7.3, 95 mM NaCl, 1.8 mM CaCl$_2$, 0.7 mM MgCl$_2$, and 100 mM mannitol was initially perfused and after 1 min was switched to the hypotonic extracellular solution containing 10 mM HEPES-NMDG pH 7.3, 95 mM NaCl, 1.8 mM CaCl$_2$, and 0.7 mM MgCl$_2$. The intracellular solution contained 10 mM HEPES-NMDG pH 7.3, 150 mM NMDG-Cl, 1 mM EGTA, and 2 mM Na-ATP. For experiments investigating Ca$^{2+}$ activation, the intracellular solution additionally contained 1 mM of free Ca$^{2+}$. The osmolality of the solutions was determined with a vapor pressure osmometer (VAPRO) to be 301 mmol kg$^{-1}$ for the isotonic extracellular solution, 202 mmol kg$^{-1}$ for the hypotonic extracellular solution, and 295 mmol kg$^{-1}$ for the intracellular solution. After establishment of the whole-cell configuration, currents were monitored using a ramp protocol delivered every 2 s for 5–8 min.

For investigation of the proton channel function of TTYH proteins, we used the same experimental set-up but with solutions containing 80 mM MES-NMDG pH 7.2 (standard pH) or pH 5.5 (low pH), 1 mM EGTA, and 100 mM mannitol. For designing the composition of solutions, we were guided by previously established electrophysiological assays[31] and used HPC to probe activation by swelling. The osmolarity was measured to be 248 mmol kg$^{-1}$ for the standard-pH solution and 197 mmol kg$^{-1}$ for the low-pH solution. During recordings, cells were initially perfused with the standard-pH solution. After 1 min, the outside solution was changed to the low-pH solution. The standard-pH solution was used as pipette solution for recordings in the absence of Ca$^{2+}$. For recordings at elevated calcium, the standard-pH solution was supplemented with 1 mM free Ca$^{2+}$. Pipette resistance was typically 18–25 MΩ when filled with the intracellular solution. Series resistance was typically of 15–25 MΩ. After establishment of the whole-cell configuration, currents were monitored using a ramp protocol delivered every 2 s for 5–8 min. All data were analyzed in Clampfit 10.7 (Molecular Devices) and Microsoft Excel. Electrophysiology data are accessible via the Dryad database[59].

**Statistics and reproducibility**. Electrophysiological recordings were repeated multiple times from different transfections with very similar results. Conclusions of experiments were not changed upon inclusion of further data. In all cases, leaky patches were discarded.

**Reporting summary**. Further information on research design is available in the Nature Research Reporting Summary linked to this article.

## Data availability
The data that support this study are available from the corresponding authors upon reasonable request. The three-dimensional cryo-EM density maps of full-length TTYH2, TTYH2 in nanodiscs, TTYH1, and TTYH3 have been deposited in the Electron Microscopy Data Bank under accession numbers EMD-13194, EMD-13201, EMD-13200, and EMD-13198. Each deposition includes a corresponding full map, both half-maps and the mask used for final FSC calculation. Coordinates for the models of the full-length TTYH2, TTYH2 in nanodiscs, TTYH1, and TTYH3 have been deposited in the Protein Data Bank under accession numbers 7P54, 7P5M, 75PJ, and 7P5C, respectively. The data from electrophysiological recordings and the liposome assay showing the absence of chloride and proton conduction in TTYH proteins have been deposited in the Dryad database (https://doi.org/10.5061/dryad.fn2z34ttx). The data from the scrambling assay and lipidomics of TTYH2 have been deposited in the Dryad database (https://doi.org/10.5061/dryad.69p8cz92n). Source data are provided with this paper.

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

## Acknowledgements

This research was supported by grants from the Swiss National Science Foundation (No. 31003A_163421 to R.D.). Simona Sorrentino and the Center for Microscopy and Image Analysis (ZMB) of the University of Zurich are acknowledged for the support and access to the electron microscope. The cryo-electron microscope and K3-camera were acquired with support of the Baumgarten and Schwyzer-Winiker foundations and a Requip grant of the Swiss National Science Foundation. We thank T. J. Jentsch for providing the *LRRC8*−/− HEK cell line and S. Rutz for the preparation of initial expression constructs. Lipid analysis was performed with the help of the FGCZ of UZH/ETH Zurich. The support of Sebastian Streb is acknowledged. All members of the Dutzler laboratory are acknowledged for their help at various stages of the project.

## Author contributions

A.S. generated expression constructs, purified proteins, and performed electrophysiology and flux experiments. M.S. oversaw cryo-EM experiments and prepared the samples for cryo-EM. M.S. and M.S.S. collected cryo-EM data. A.S. and M.S. proceeded with structure determination and refinement. D.D. provided intellectual input throughout. A.S., M.S., and R.D. jointly planned experiments, analyzed the data, and wrote the manuscript.

## Competing interests

The authors declare no competing interests.
