## [Peer Review File · Nature Communications]

Reviewers' Comments:

Reviewer #1:

Remarks to the Author:

Cryo-EM structures of the TTYH family reveal a novel architecture for lipid interactions.

In this joint study by the Dutzler and Sawicka groups, they performed cryo-EM structural and in-depth functional studies to delve into the functional role of the TTYH family proteins. Contrary to previous studies by other groups, they found that this protein family is neither volume-regulated nor Ca²⁺-gated anion channel. Based on the unique structural feature of this protein family, they speculated a functional role of this protein family in lipid or hydrophobic molecule binding.

This reviewer believes that this is an important study, as it clearly demonstrated that previous roles of this family of proteins in volume or Ca²⁺-regulated anion channels are incorrect. Their structural and functional data are convincing. They also tested the role of this protein in H⁺ permeation, which they could not detect. The papers are mostly clearly written. Both structural and functional studies were done well. Therefore, I recommend publication of this study in Nature Communications after they address my suggestions.

Major comments

1) I strongly suggest that authors change the title to something more meaningful. To me, the facts that TTYH having a novel fold and its potential role in lipid interactions are secondary to the findings that both structural and functional studies of TTYH strongly debunk the previous claims that the TTYH is an anion channel. I think this is the most important take-home message from this manuscript. I suggest authors to change the title something like "Structural and functional analyses reveal that the TTYH family is not an anion channel".

2) Structure modeling: authors need to re-refine their models.

The high-resolution structural model (Ttyh3 GDN, 3.2 angstrom) exhibits the highest clash score >10, highest molprobity score 2.52, highest rotamer outlier (~7.7%). The provided cryo-EM density looks decent, so this has to do with modeling. Also, side chain outlier should be all below 1% for all other models as well. I suggest that authors re-refine the models for both Ttyh2 ND and Ttyh2 GDN.

Minor comments

1) Have authors tested H⁺ permeations under high concentration of cytosolic Ca²⁺? The structure indicates at least it is more suitable for H⁺ channel than anion channel.

2) Line 132, authors mention "disulfide bond in in Fig. 2d", but I don't see the disulfide bond in Fig. 2d.

3) Page 11, lines 219-224. The sentence needs to be re-written. It is difficult for me to follow what the authors are trying to say. Also Extended Data Fig. 7c is likely a wrong figure reference here.

4) Extended Data Figs. 3-5: similar colors were used for the cryo-EM density maps and the models so that it is difficult to assess how well the models fit the respective density maps. I suggest that authors change the color of the density maps to a lighter color.

Reviewer #2:

Remarks to the Author:

This manuscript describes novel folded structures of three TTYH paralogs, whose function has long been predicted as Ca²⁺- or swelling-gated chloride channel, especially in neural cells in brain. In this paper, the authors present the cryo-EM structures of TTYH1-3 structures, which shows novel architecture with 5 TMs and long-standing extracellular domains, which have a role in dimer formation. While the structure is so unusual, there seems no channel to conduct ions, which was confirmed by the authors using electrophysiological analyses. Instead, they found elongated density binding to the long and wide cavity inside the proteins, which proposes a possibility that TTYH

functions in lipid transport, still remaining functional possibilities of signal transduction or membrane enzymes. The structural analyses are solid, and the paper is well written for this reviewer to read easily. However, this reviewer would like to claim several concerns, as follows.

1. While the authors carried out the extraction and liquid chromatography/mass spectrometry of included lipids, this reviewer would like to ask the authors to conduct liposome-based analysis of lipid transport to uncover the exact function of TTYH proteins. The authors are specialists of biochemistry of membrane scramblase (eLife, 2019), this experiment seems feasible.

2. In Fig. 5, all the three TTYHs bind lipids at the hydrophobic cavity, where the hydrophilic phosphate head groups are exposed to the solvent. If TTYHs act as membrane scramblase, are the hydrophilic head groups exposed to hydrophobic acyl chains in the membrane? Authors should execute.

3. In Fig. 1, TM1 and TM2 should be connected by a dotted line to clarify the protein architecture.

4. In Fig. 2d, a disulfide bond to stabilize the extracellular flap is not shown.

We thank the reviewers for their constructive comments, which we have considered in our revised manuscript and which we address in detail below.

Reviewer #1 (Remarks to the Author):

Cryo-EM structures of the TTYH family reveal a novel architecture for lipid interactions.

In this joint study by the Dutzler and Sawicka groups, they performed cryo-EM structural and in-depth functional studies to delve into the functional role of the TTYH family proteins. Contrary to previous studies by other groups, they found that this protein family is neither volume-regulated nor Ca²⁺-gated anion channel. Based on the unique structural feature of this protein family, they speculated a functional role of this protein family in lipid or hydrophobic molecule binding.

This reviewer believes that this is an important study, as it clearly demonstrated that previous roles on this family of protein in volume or Ca²⁺-regulated anion channels are incorrect. Their structural and functional data are convincing. They also tested the role of this protein in H⁺ permeation, which they could not detect. The papers are mostly clearly written. Both structural and functional studies were done well. Therefore, I recommend publication of this study in Nature Communications after they address my suggestions.

Major comments

1) I strongly suggest that authors change the title to something more meaningful. To me, the facts that TTYH having a novel fold and its potential role in lipid interactions are secondary to the findings that both structural and functional studies of TTYH strongly debunk the previous claims that the TTYH is an anion channel. I think this is the most important take-home message from this manuscript. I suggest authors to change the title something like “Structural and functional analyses reveal that the TTYH family is not an anion channel”.

We appreciate this suggestion of the reviewer but prefer to refrain from including a negative result in the title. Although the proposed title might sound appealing at this point, we hope that the manuscript will make a lasting impact in defining the architecture of a novel membrane protein family of currently unknown function. Functional properties of poorly characterized proteins are frequently annotated incorrectly, a problem that is particularly prevalent in the chloride channel field. The initial misannotation is usually clarified at one point, after which the field moves on to explore the actual role of the respective proteins. After considering alternatives

we would prefer to stick with our original title since we think that the interaction with lipids points towards the actual function of TTYH proteins.

2) Structure modeling: authors need to re-refine their models.

The high-resolution structural model (Ttyh3 GDN, 3.2 angstrom) exhibits the highest clash score >10, highest molprobability score 2.52, highest rotamer outlier (~7.7%). The provided cryo-EM density looks decent, so this has to do with modeling. Also, side chain outlier should be all below 1% for all other models as well. I suggest that authors re-refine the models for both Ttyh2 ND and Ttyh2 GDN.

We have corrected the side-chain outliers and improved the structures of TTYH2 and 3. This is reflected in the improved statistics in Table 1. We also have added FSC plots of refined models against the cryo-EM full and half-maps for model validation (displayed in Supplementary Fig. 6).

Minor comments

1) *Have authors tested H⁺ permeations under high concentration of cytosolic Ca²⁺? The structure indicates at least it is more suitable for H⁺ channel than anion channel.*

We have now included novel data where we investigate H⁺ permeation at elevated (1 mM) intracellular Ca²⁺ concentrations and display them as Supplementary Fig. 9f-j. Similar to equivalent data in the absence of Ca²⁺ (Supplementary Fig. 9a-e), we did not detect any activity pointing towards H⁺ transport.

2) *Line 132, authors mention “disulfide bond in in Fig. 2d”, but I don’t see the disulfide bond in Fig. 2d.*

We have revised Fig. 2d. and highlighted both disulfide bridges to indicate their location more clearly.

3) *Page 11, lines 219-224. The sentence needs to be re-written. It is difficult for me to follow what the authors are trying to say. Also Extended Data. Fig. 7c is likely a wrong figure reference here.*

We have rewritten the sentence to make it easier to comprehend. Additionally, the entire paragraph was restructured to introduce the lipid scrambling data.

Line 218-226:

Since its membrane-embedded inner boundary is delimited by charged and hydrophilic residues reminiscent of a lipid scramblase³⁴ (Fig. 3a), we were interested whether TTYH proteins would facilitate the movement of lipids between both leaflets of the bilayer. We have investigated this question by employing an assay that was previously used to characterize the functional properties of lipid scramblases of the TMEM16 family³⁴⁻³⁶. For that purpose, we have reconstituted TTYH2 into liposomes containing traces of fluorescently labeled lipids and assayed the bleaching of lipids residing in the outer leaflet in absence and presence of Ca²⁺. However, in no case did we find any evidence for TTYH2 to facilitate lipid flip-flop (Supplementary Fig. 11).

4) Extended Data Figs. 3-5: similar colors were used for the cryo-EM density maps and the models so that it is difficult to assess how well the models fit the respective density maps. I suggest that author change the color of the density maps to a lighter color.

We have changed the coloring of the maps displayed in Supplementary Figs. 2-5 to lighter colors to better illustrate the fits.

Reviewer #2 (Remarks to the Author):

This manuscript describes novel folded structures of three TTYH paralogs, whose function has long been predicted as Ca²⁺- or swelling-gated chloride channel, especially in neural cells in brain. In this paper, the authors present the cryo-EM structures of TTYH1-3 structures, which shows novel architecture with 5 TMs and long-standing extracellular domains, which have a role in dimer formation. While the structure is so unusual, there seems no channel to conduct ions, which was confirmed by the authors using electrophysiological analyses. Instead, they found elongated density binding to the long and wide cavity inside the proteins, which proposes a possibility that TTYH functions in lipid transport, still remaining functional possibilities of signal transduction or membrane enzymes. The structural analyses are solid, and the paper is well written for this reviewer to read easily. However, this reviewer would like to claim several concerns, as follows.

1. While the authors carried out the extraction and liquid chromatography/mass spectrometry of included lipids, this reviewer would like to ask the authors to conduct liposome-based analysis of lipid transport to uncover the exact function of TTYH proteins. The authors are specialists of biochemistry of membrane scramblase (eLife, 2019), this experiment seems feasible.

We have now included data that investigate a potential catalytic activity of TTYH2 in facilitating the movement of lipids between both leaflets of the bilayer (shown in Supplementary Fig. 10). Neither in absence nor presence of Ca^{2+} , we were able to detect any activity as lipid scramblase.

2. In Fig. 5, all the three TTYHs bind lipids at the hydrophobic cavity, where the hydrophilic phosphate head groups are exposed to the solvent. If TTYHs act as membrane scramblase, is the hydrophilic head groups are exposed to hydrophobic acyl chains in the membrane? Authors should execute.

It should be emphasized that the density inside the cavity is not of sufficient resolution to assign the chemical nature of the interacting molecules. The interpretation is thus to some degree arbitrary but takes the physico-chemical properties of the cavity into account. This cavity is for most of its part located outside of the membrane facing the aqueous environment. Since its interior is highly hydrophobic, the orientation of modeled fatty acids and cholesterol molecules was assigned so that the hydrophobic part faces the inside of the cavity whereas the hydrophilic headgroups point towards the aqueous surrounding to optimize their interaction with the respective environment. As shown in Supplementary Fig. 10, we do not find any evidence for TTYH proteins functioning as lipid scramblase facilitating the movement of lipids between both leaflets of the membrane. Instead we envision that TTYH might facilitate the extraction of hydrophobic and amphiphilic molecules embedded in the bilayer from the membrane.

3. In Fig. 1, TM1 and TM2 should be connected by dotted line to clarify the protein architecture.

We have introduced the requested change to the figure.

4. In Fig. 2d, disulfide bond to stabilize the extracellular flap is not shown.

We have revised Fig. 2d. to better display the two disulfide bridges.